

# Length-mass relationships of pond macroinvertebrates do not hold between Southern and Northern Europe

Vladimíra Dekanová[1], Marek Svitok[2,3,4], Sara Bento[1], João Caramelo[1], Pedro Peixe e Sousa[1] and Bruno M. Carreira[1]

[1] cE3c - Centre for Ecology, Evolution and Environmental Changes & CHANGE – Global Change and Sustainability Institute, Faculty of Sciences, University of Lisbon, Lisbon, Portugal
[2] Department of Biology and General Ecology, Faculty of Ecology and Environmental Sciences, Technical University in Zvolen, Zvolen, Slovakia
[3] Department of Forest Ecology, Faculty of Forestry and Wood Sciences, Czech University of Life Sciences Prague, Prague, Czech Republic
[4] Institute of Botany, Plant Science and Biodiversity Centre, Slovak Academy of Sciences, Bratislava, Slovakia

Corresponding author
Vladimíra Dekanová,
dekanovav@gmail.com

## ABSTRACT

The lack of reliable data on length-mass relationships, essential to obtain accurate biomass estimates, limits our ability to easily assess secondary production by aquatic invertebrates. In the absence of published equations from similar habitat conditions, authors often borrow equations developed in geographic regions with different climate conditions, which may bias biomass estimates. A literature overview of published size-mass relationships for Portugal and Sweden highlights the need for further data within these biogeographic regions. We increased the number of equations available to Southern and Northern Europe, developing 18 new length-mass relationships for two families and 10 genera in Portugal and Sweden. All equations were published for the first time for these countries, except the genus *Asellus*. Our length-mass relationships were obtained from specimens collected on a one-time sampling of eight ponds in Portugal and five ponds in Sweden during late spring in 2023. Dry mass (DM) was modelled as a function of body length (BL), using the natural log-linear function with a power model ($\ln DM = \ln a + b \times \ln BL$). The equations obtained were compared with linear mixed models testing the fixed effects of "body length" and "country", as well as their interaction. A comparison of the equations developed in this study showed country-specific differences for all taxa, expect the genus *Caenis*, indicating a low potential transferability of the equations between Southern and Northern Europe. In contrast, the comparison of the equation obtained for *A. aquaticus* in this study with an equation published for this taxon in Sweden showed great similarities, suggesting a high transferability. Recommending caution in the borrowing of published length-mass equations, that can differ drastically between different geographic and climatic regions, especially at larger sizes, we provide a series of guidelines and good practices in this field.

## INTRODUCTION

The ability to quantify secondary production is crucial to trace energy flow and understand the dynamics of ecosystems. The secondary production of invertebrates is critical in the characterization of energy transfer in aquatic ecosystems (*Benke et al., 1999*; *Huryn & Wallace, 2000*). Invertebrates play an important role in the functioning of aquatic ecosystems, greatly contributing to nutrient cycling, especially in ponds (*Fehlinger et al., 2023*), often dominating aquatic food webs (*Habdija, Lajtner & Belinić, 1995*; *Wallace & Webster, 1996*; *Huryn & Wallace, 2000*; *Hauer & Resh, 2017*). Thus, quantifying their production constitutes a powerful tool to study community structure, seasonal patterns, and resource distribution (*Rigler & Downing, 1984*; *Benke, 1993*).

The reliance on the determination of macroinvertebrate biomass for the assessment of secondary productivity in aquatic ecosystems stresses the importance of obtaining accurate estimates (*Walther et al., 2006*; *Martin, Proulx & Magnan, 2014*). Macroinvertebrate biomass is usually obtained following one of three methods: direct weighing of fresh (as wet weight) or preserved animals (as dry weight); estimation of biovolume; or size-mass conversion (*Burgherr & Meyer, 1997*). Because weighing aquatic macroinvertebrates in the field is often impractical, the latter method is the most commonly employed (*Johnston & Cunjak, 1999*; *González, Basaguren & Pozo, 2002*). The samples are fixed immediately after collection and biomass is estimated indirectly using length-mass relationships derived from size and weight measurements of the preserved organisms (*Johnston & Cunjak, 1999*; *Wetzel, Leuchs & Koop, 2005*; *Benke & Huryn, 2007*). This approach is faster, more precise and time-efficient than the other methods. Furthermore, it can be used on invertebrates of all sizes, does not destroy the specimens' bodies (*Burgherr & Meyer, 1997*; *Johnston & Cunjak, 1999*) and, due to its low effort, it allows for more comprehensive comparisons of invertebrate populations between different habitats and ecosystems (*Benke, 1993*).

Despite the great progress in the last fifty years (*e.g.*, *Smock, 1980*; *Benke et al., 1999*; *Johnston & Cunjak, 1999*), equations describing aquatic macroinvertebrates size-mass relationships are still largely missing for many species, especially for insects of the orders Megaloptera, Hemiptera and Odonata. Moreover, environmental conditions, such as water temperature (*Schröder, 1987*) and chemistry (*Meyer, 1989*; *Burgherr & Meyer, 1997*), and food availability can affect size-mass relationships (*Gee, 1988*; *Basset & Glazier, 1995*), potentially leading to substantial differences among populations of the same species (*Johnston & Cunjak, 1999*). A consequence of this intraspecific variation, researchers often need to develop their own length-mass relationships to avoid inaccurate biomass estimations (*Walther et al., 2006*; *Martin, Proulx & Magnan, 2014*), and error propagation to ecosystem metrics (*e.g.*, biomass, secondary production).

While developing size-mass relationships from actual samples is preferable (*Smock, 1980*; *González, Basaguren & Pozo, 2002*), borrowing equations from the literature is a rather common practice. Nevertheless, it should be done with caution. Studies comparing biomass estimates obtained for aquatic macroinvertebrates with developed and borrowed equations are still rare and the potential transferability of the size-mass relationships can be hard to predict. While some studies argue that geographic distance and climatic

conditions can considerably affect length-mass relationships (*Hajiesmaeili, Ayyoubzadeh & Abdoli, 2019*; *Dekanová, Venarsky & Bunn, 2022*), other authors claim that length-mass models can be used in different ecosystems, provided that the environmental conditions are similar (*Rosati, Barbone & Basset, 2012*). Temperature is an environmental variable of paramount importance, as it is one of the main drivers of growth rate in aquatic invertebrate larvae (*Brittain, 1983*; *Bonacina et al., 2023*). According to the temperature-size rule (*Atkinson, 1995*), differences in environmental temperature should be reflected in adult size, with higher temperatures leading to smaller sizes, which may also affect the size-mass relationships of aquatic invertebrates.

The size-mass relationships of aquatic macroinvertebrates are better studied in Europe than in other parts of the world. However, the available equations are still relatively few, scattered in the literature, and do not provide equal coverage of all taxa or climatic regions across the continent. Here, we present an overview of the published length-mass relationships of aquatic macroinvertebrates from two European countries at latitudinal extremes (Portugal and Sweden) and expand the number of equations available for pond macroinvertebrates in Europe, by developing 18 new length-mass relationships for taxa from five orders (Isopoda, Ephemeroptera, Odonata, Hemiptera, and Coleoptera). Furthermore, we assessed the potential transferability of the length-mass relationships developed in the two regions, by comparing the equations obtained for the same taxa in Portugal and Sweden. We hypothesised that the geographical distance between our study areas in Southern and Northern Europe, together with the contrasting climate conditions and environmental constraints in these two regions, should affect the taxa surveyed in this study and lead to differences in the length-mass relationships of the pond macroinvertebrates. Hence, we predicted that the potential transferability of the equations obtained for the same taxa in the two study areas should be low between Portugal and Sweden. Lastly, we compared the equations developed in this study with those published in the literature for the same taxa, predicting a high potential transferability between equations obtained within the same region.

## MATERIALS AND METHODS

### Overview of published length-mass relationships

We assessed the number of published length-mass relationships available in the literature for the freshwater communities of Portugal and Sweden, by performing a systematic search on the Google Scholar database in October 2023. For this, we used the following search terms: "length mass" OR "length weight" OR "size mass" OR "size weight" "relationship" AND "aquatic invertebrate" AND "Portugal" OR "Sweden".

### Study areas and focal taxa

We determined length-mass relationships for macroinvertebrate taxa present in ponds from Southwestern and Northern Europe (Fig. 1, Table 1). Situated in southwestern Europe, southwest Portugal is characterised by a termo-Mediterranean climate, with warm arid summers and cold windy winters; with air temperature typically ranging from 7 °C to
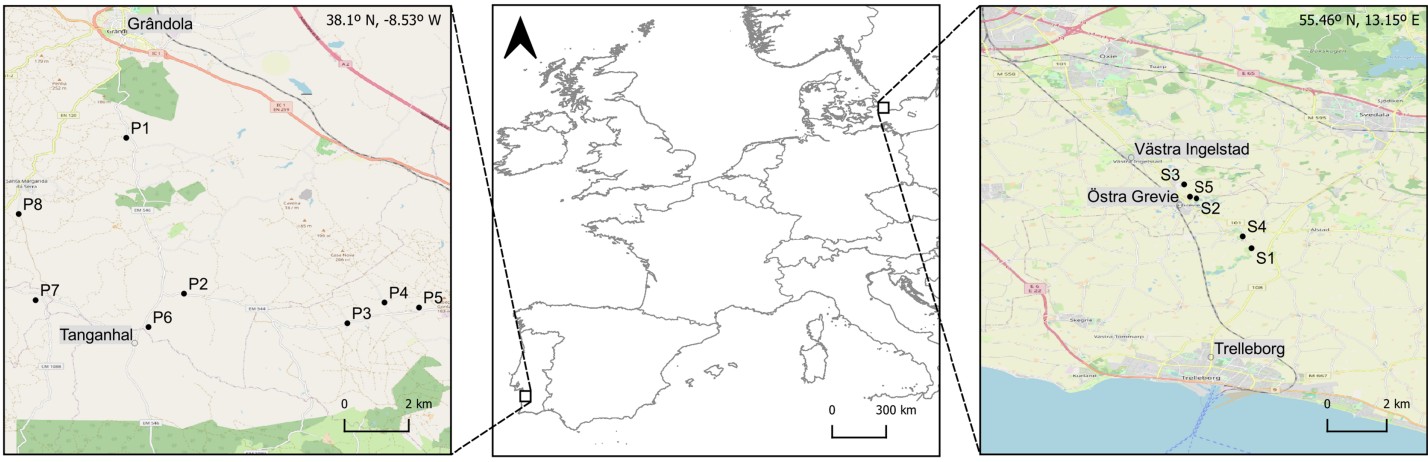

**Figure 1 Geographic location of the ponds surveyed in the two study areas in Southern (Grândola, Portugal; *n* = 8) and Northern Europe (Trelleborg, Sweden; *n* = 5).** Maps of the study areas are based on *OpenStreetMap Contributers (2021)*.

**Table 1 Pond characteristics.**

| Country | Pond | Latitude (°N) | Longitude (°E) | Altitude (m) | Surface area (m²) | Water depth (m) | Water temperature (°C) | Conductivity (µS.cm⁻¹) | Oxygen concentration (mg.L⁻¹) | pH |
|---|---|---|---|---|---|---|---|---|---|---|
| Portugal | P1 | 38.138753 | −8.566047 | 116 | 1,025 | 0.44 | 27.4 | 312 | 14.04 | 9.03 |
| | P2 | 38.083453 | −8.545456 | 219 | 434 | 0.54 | 25.7 | 347 | 6.04 | 7.35 |
| | P3 | 38.073022 | −8.487406 | 122 | 50 | 0.40 | 22.0 | 189 | 6.09 | 7.75 |
| | P4 | 38.080325 | −8.474219 | 137 | 3,420 | 0.49 | 20.8 | 180 | 2.84 | 6.96 |
| | P5 | 38.078542 | −8.461964 | 139 | 756 | 0.80 | 24.3 | 314 | 7.43 | 7.67 |
| | P6 | 38.071647 | −8.558103 | 223 | 320 | 0.46 | 25.7 | 612 | 16.08 | 9.97 |
| | P7 | 38.081117 | −8.597997 | 250 | 510 | 0.50 | 24.0 | 459 | 5.77 | 7.49 |
| | P8 | 38.111817 | −8.604150 | 253 | 350 | 0.55 | 19.1 | 187 | 7.30 | 7.75 |
| Sweden | S1 | 55.444424 | 13.173338 | 23 | 2,000 | 0.80 | 20.5 | 606 | 3.91 | 7.58 |
| | S2 | 55.471082 | 13.144300 | 44 | 8,400 | 0.13 | 17.9 | 731 | 1.55 | 6.95 |
| | S3 | 55.478354 | 13.137272 | 37 | 240 | 0.86 | 18.7 | 460 | 2.88 | 6.80 |
| | S4 | 55.450990 | 13.168690 | 36 | 1,650 | 0.40 | 22.3 | 417 | 9.14 | 7.87 |
| | S5 | 55.471975 | 13.140226 | 23 | 1,200 | 0.70 | 22.2 | NA | NA | 8.07 |

**Note:**

Summary of the GPS coordinates and main environmental characteristics of the 13 ponds sampled in Portugal and Sweden (see Fig. 1).

30 °C during the year. Situated in northern Europe, southern Sweden is characterised by a temperate oceanic (Baltic) climate, with partly cloudy summers and long, very cold, snowy, windy winters; with air temperature typically ranging from −2 °C to 21 °C during the year. The geographical separation between the study areas is large (over 3,000 km and 17° of latitude), and, consequently, the pond communities surveyed experience different climate conditions and environmental constraints. In Southern Europe, the life cycle of pond macroinvertebrates is mainly constrained by desiccation and high water temperatures,

**Table 2 Length-mass relationships developed for the pond macroinvertebrates collected in Portugal and Sweden.**

| Order | Family | Genus | Country | n | Range (mm) | ln a | SE | b | SE |
|---|---|---|---|---|---|---|---|---|---|
| Isopoda | Asellidae | Asellus | Sweden | 211 | 2.4–16.0 | −5.2742 | 0.1100 | 2.9704 | 0.0465 |
| Ephemeroptera | Baetidae | Cloeon | Portugal | 135 | 2.9–9.0 | −3.7406 | 0.1763 | 1.8299 | 0.0955 |
| | | | Sweden | 129 | 5.1–11.6 | −7.5214 | 0.3526 | 3.5890 | 0.1651 |
| | Caenidae | Caenis | Portugal | 45 | 2.9–7.0 | −4.1558 | 0.6761 | 2.0809 | 0.3946 |
| | | | Sweden | 114 | 2.3–8.8 | −4.4305 | 0.6258 | 2.1238 | 0.3194 |
| Odonata | Coenagrionidae | | Portugal | 300 | 2.3–21.9 | −5.0348 | 0.3286 | 2.3374 | 0.1193 |
| | | | Sweden | 187 | 6.4–23.8 | −7.2736 | 0.4309 | 3.1407 | 0.1569 |
| | Aeshnidae | Anax | Portugal | 55 | 5.2–33.5 | −5.2473 | 0.4048 | 2.7703 | 0.1340 |
| | | Aeshna | Sweden | 33 | 4.3–23.3 | −5.5252 | 0.3922 | 2.5385 | 0.1652 |
| | Libellulidae | Sympetrum | Portugal | 97 | 5.6–19.2 | −6.4656 | 0.3650 | 3.3861 | 0.1348 |
| | | | Sweden | 83 | 3.8–16.2 | −3.8864 | 0.3647 | 2.1025 | 0.1555 |
| Hemiptera | Corixidae | Corixa | Portugal | 46 | 3–7.2 | −2.9446 | 0.2910 | 1.9940 | 0.1796 |
| | | Sigara | Sweden | 77 | 1.5–8.3 | −4.3306 | 0.1353 | 2.5705 | 0.0882 |
| | Notonectidae | Notonecta | Portugal | 195 | 2.3–16.5 | −3.8855 | 0.2476 | 2.7167 | 0.1228 |
| | | | Sweden | 58 | 2.4–17.5 | −4.5538 | 0.3054 | 2.9085 | 0.1543 |
| | Naucoridae | Naucoris | Portugal | 66 | 1.2–10.0 | −3.2901 | 0.2349 | 2.4167 | 0.1360 |
| Coleoptera | Dytiscidae | | Portugal | 81 | 2.7–44.3 | −2.1355 | 0.3345 | 0.9988 | 0.1724 |
| | | | Sweden | 86 | 7.6–40.0 | −4.0859 | 0.6517 | 1.8232 | 0.2980 |

**Note:**
We show the parameters of the power model (ln DM = ln $a$ + $b$ × ln BL) of the relationships between body length (BL (mm)) and dry mass (DM (mg)), where $n$, sample size; Range, size range of body length (mm); $a$ and $b$, fitted regression constants; and SE, standard error.

while in Northern Europe it is mainly constrained by icing and long periods of low water temperatures (*Cedar Lake Ventures, Inc., 2019*).

We obtained 18 new length-mass relationships for two families and seven genera of freshwater macroinvertebrates from Portugal ($n$ = 1,020 specimens) and Sweden ($n$ = 978 specimens) that were sufficiently abundant to develop the equations (see Table 2). While Portugal and Sweden shared most of the surveyed taxa, especially common and widespread taxa, there were a few exceptions. We developed a length-mass equation for the genus *Asellus*, a common shredder of the Isopoda order with dorsoventrally flattened body distributed throughout the temperate zone in Europe (*Verovnik, Sket & Trontelj, 2005*), that was only found in Sweden. We developed length-mass equations for two genera of gatherer collectors of the Ephemeroptera order, the cosmopolitan genus *Cloeon*, which is usually the most common mayfly in ponds; and the small sized genus *Caenis*, which occurs in high densities on the bottoms of shallow ponds (*Bauernfeind & Soldan, 2013*). Furthermore, we developed length-mass equations for predatory taxa from the orders Hemiptera (four genera), Odonata and Coleoptera. The taxa from the order Hemiptera comprised the genus *Notonecta*, with strongly dorsally convex and ventrally flattened body, which occurs in western and central Europe; the genera *Corix* and *Sigara* from the family Corixidae, which are predators or macrophyte piercers with flattened body that use the hind legs for swimming and are found in a wide range of habitats, often as the most common insect (*Schuh & Slater, 1995*); and the genus *Naucoris*, characteristic of well

vegetated stagnant waters and mainly distributed in tropical and subtropical regions (*Zettel, Nieser & Polhemus, 1999*), that was only found in Portugal. The taxa from the order Odonata comprised the family Coenagrionidae, characterized by its elongated abdomen, which is the most widely distributed family of the suborder Zygoptera in Europe (*De Knijf et al., 2024*); two genera of the family Aeshnidae, *Aeshna* and *Anax*, which are insectivore taxa with similar morphology and occur in well vegetated still waters of southern and central Europe; and the genus *Sympetrum*, which is a small to medium-sized dragonfly commonly found across Europe amongst gravel, vegetation, detritus and mud (*Dolný, Harabiš & Bárta, 2016*). The overview of family, genus and species distribution of the surveyed taxa was based on the online databases *Naturdata (2012)* (Portugal) and Biodiversity4all (*Tiago, 2020*) (Sweden).

## Macroinvertebrate collection

In southwestern Europe, macroinvertebrates were collected during spring, between the 1st and the 4th of May 2023, in the rural surroundings of Grândola (Portugal, 38°N), in eight ponds within a 6 km radius (Fig. 1, Table 1). In northern Europe, macroinvertebrates were collected during late spring, between the 1st and the 7th of June 2022, in the rural surroundings of Trelleborg (Sweden, 55°N), in five ponds within a 2 km radius (Fig. 1, Table 1). To cover the diversity of macroinvertebrate taxa and the natural variation in their body mass and length, sampling was conducted in all the available microhabitats and at different depths within each pond. We collected individuals from the orders Isopoda (genus *Asellus*), Ephemeroptera (genera *Caenis*, *Cloeon*), Hemiptera (genera *Corixa*, *Sigara*, *Notonecta*, *Naucoris*), Odonata (genera *Sympetrum*, *Anax*, *Aeshna* and family Coenagrionidae) and Coleoptera (family Dytiscidae). Individuals were collected with a D-shaped net (200-µm mesh size) using kick sampling (*Wetzel & Likens, 2000*). This method was supplemented with hand-collecting and sweeping with a kitchen strainer (19 cm in diameter; 1 mm mesh). Immediately after collection in the field, the individuals were preserved in a solution of 96% ethanol and transferred to plastic bottles (500 mL) for further processing in the laboratory. In each pond, we recorded water parameters— temperature (°C), pH, conductivity ($\mu S.cm^{-1}$) and dissolved oxygen ($mg.L^{-1}$)—in three replicates (Table 1), using a portable YSI multimeter (model 6909 ProDSS; YSI Environmental, Yellow Springs, OH, USA).

## Sample processing

Macroinvertebrates were identified using the keys developed by *Jansson (1969)*, *Rozkošný (1980)*, *Tachet et al. (2010)*, *Dolný, Harabiš & Bárta (2016)* and *Kriska (2022)*. When genus-level identification was not possible, or the number of individuals of the genus was insufficient to estimate a length-mass relationship ($n < 10$, *Johnston & Cunjak, 1999*), individuals were grouped at the family level. The individuals of each taxon were sorted and separated into different size classes that consisted of 1 mm increments from the smallest to the largest body length. Then, one to three individuals were measured per size class, *i.e.*, per 1 mm increment in body size, recording the body length of larval stages for all macroinvertebrate taxa, including full-grown larvae with black wing pads for

Ephemeroptera. We also recorded the body length of adults of the order Hemiptera, as both larval stages and adults of this taxonomic group share similar morphological features and preferred habitats. We measured body length, a widely used morphometric trait in length-mass relationships in aquatic invertebrates, since its broader range provides higher determination coefficients than head width (*Rosati, Barbone & Basset, 2012*). The body length was defined as the distance from the frontal part of the head capsule to the posterior of the last abdominal segment, excluding the cerci and anal prolegs. Measurements were taken with a stereomicroscope (model SMZ800; Nikon, Tokyo, Japan) with an ocular micrometre of 0.05 mm accuracy (see Table S1). Overall, the number of individuals per size class was evenly distributed for most taxa, with two exceptions. The size distribution of the order Hemiptera was skewed towards small and medium-sized individuals, with only a small number of large adults included in the models (*n* = 76). Additionally, the size distribution of the family Dytiscidae displayed a few gaps in the size classes, that may have resulted from differences in the size ranges of the four subfamilies (five genera) composing this family.

After measuring total length, each individual was transferred into a pre-weighed aluminium cup, that was subsequently oven-dried at 60 °C for 24 h following standard protocol (*Benke & Huryn, 2007*). Then, their dry mass (DM) was determined to the nearest 0.01 mg on a high precision scale (model PX224M; Ohaus Pioneer, Zürich, Switzerland) (see Table S1). The number of individuals used to estimate the length-mass relationship for each taxon ranged between 33 for *Aeshna* and 300 for Coenagrionidae, both in Portugal (Table 2).

## Statistical analysis

Dry mass (DM) was modelled as a function of body length (BL) using a natural log-linear function with a power model (ln DM = ln $a$ + $b$ × ln BL), where $a$ and $b$ represent the coefficients of the intercept and the slope of the relationship, respectively. We used this power model as it is the most commonly employed in studies developing length-mass relationships (*Burgherr & Meyer, 1997*; *Johnston & Cunjak, 1999*). We developed equations at the genus level for all surveyed taxa, except for two equations at the family level (Coenagrionidae and Dytiscidae). This approach allowed us to maximize the number of comparisons between countries, while undertaking an acceptable compromise in the accuracy of our models. Equations at the genus or family level are less accurate than equations at the species level, but it is common to find length-mass relationships at the genus level in the published literature (*e.g.*, *Smock, 1980*; *Burgherr & Meyer, 1997*; *Oliphant & Hyslop, 2020*; *Dekanová, Venarsky & Bunn, 2022*). Moreover, a study by *Giustini et al. (2008)* showed small differences between estimated and measured biomass at the genus level in the genera *Leuctra* (Plecoptera) and *Baetis* (Ephemeroptera), suggesting that species share similar biomass-size growth patterns within a given genus.

We employed linear mixed-effects models (LMMs; *Pinheiro & Bates, 2006*) to compare the length-mass relationships. Each model included the fixed effects of body length and country of origin, as well as their interaction, to test for potential differences between Portugal and Sweden. Simpler length-mass models without specifying country of origin

were used for two of the surveyed taxa, that were only found in either Sweden (*Asellus*) or Portugal (*Naucoris*). To account for the non-independence of data from individuals collected in the same pond, pond identity was treated as a random effect in the LMMs. The Akaike Information Criterion (AIC) was used to specify the random effect structure. Both random intercept and random slope models were fitted for each taxon and the one with the lower AIC value was further evaluated. We screened all models for homoscedasticity, linearity and normality using diagnostic plots of the residuals and no considerable violations of the model assumptions were detected (See Figs. S1 and S2). The statistical significance of the fixed effects was assessed using F tests with Satterthwaite's approximation of degrees of freedom to account for correlations among repeated measurements taken from the same ponds (*Kuznetsova, Brockhoff & Christensen, 2017*). We calculated marginal pseudo-determination coefficients ($R^2_m$; *Nakagawa, Johnson & Schielzeth, 2017*) as goodness-of-fit measures. All analyses and visualisations were performed in R version 4.2.2 (*R Core Team, 2021*) using the libraries DHARMa (*Hartig, 2022*), ggplot2 (*Wickham, 2016*), lme4 (*Bates et al., 2014*), lmerTest (*Kuznetsova, Brockhoff & Christensen, 2017*) and performance (*Lüdecke et al., 2021*).

## Comparison of the length-mass relationships developed for Portugal and Sweden

To compare the allometric slopes (*b*) of the length-mass relationships developed in this study with those published in the literature for Portugal and Sweden, we extracted the following information from each article: the taxonomic level, fitting method (linear or non-linear regression), size-mass equation (*a* and *b* constants, determination coefficient, sample size, body size range), morphometric trait measured (*e.g.*, body length, head width), type of mass measured (wet-mass, dry-mass, ash-free dry mass), type of habitat, and sample preservation method. To allow for comparison, the published equations were transformed into a log-linear equation with a power model ($\ln DM = \ln a + b \times \ln BL$) and compared visually by plotting body length (mm) *versus* dry mass (mg) data.

# RESULTS

## Literature review

For Portugal, we found 18 studies with published size-mass equations, developed for the classes Gastropoda and Bivalvia (*e.g.*, *Lillebø, Pardal & Marques, 1999*; *Gaspar, Santos & Vasconcelos, 2001*; *Vasconcelos et al., 2018*), and the orders Amphipoda (*e.g.*, *Cunha, Moreira & Sorbe, 2000*; *Pardal et al., 2000*; *Marques et al., 2003*) and Trichoptera (*e.g.*, *Canhoto, 1994*; *González & Graça, 2003*; *Azevedo-Pereira, Graça & González, 2006*). Two studies developed but did not publish the length-mass equations (*Sprung, 1994*; *França et al., 2009*), while six studies used published equations, mostly to assess spatial and temporal variation in invertebrate biomass (*e.g.*, *Maranhão et al., 2001*; *Ferreira et al., 2016*). For Sweden, only three studies published the size-mass equations, developed for the classes Gastropoda, Bivalvia, and Malacostraca—*Rumohr, Brey & Ankar (1987)* with 44 species, *Perkins et al. (2010)* with three species and *Eklöf et al. (2017)* with 14 species. A total of 17 studies developed, but did not publish, the length-mass equations

**Table 3 Similarity in the composition of the families and genera surveyed.**

| Order | Family | Genus | No. of species (Portugal) | No. of species (Sweden) | No. of species (shared) |
|---|---|---|---|---|---|
| Isopoda | Asellidae | *Asellus* | 1 | 1 | 1 |
| Ephemeroptera | Baetidae | *Cloeon* | 3 | 5 | 3 |
| | Caenidae | *Caenis* | NA | 6 | NA |
| Odonata | Coenagrionidae | | 13 | 13 | 5 |
| | Aeshnidae | *Anax* | 4 | 2 | 2 |
| | | *Aeshna* | 6 | 9 | 4 |
| | Libellulidae | *Sympetrum* | 8 | 7 | 6 |
| Hemiptera | Corixidae | *Corixa* | 4 | 1 | 0 |
| | | *Sigara* | 9 | 5 | 2 |
| | Notonectidae | *Notonecta* | 6 | 2 | 2 |
| | Naucoridae | *Naucoris* | 1 | 0 | 0 |
| Coleoptera | Dytiscidae | | 106 | 35 | 12 |

**Note:**
Number of species for each taxon in Portugal and Sweden and number of shared species between the two countries. Species distribution data taken from online databases *Naturdata (2012)* (Portugal) and Biodiversity4all (*Tiago, 2020*) (Sweden).

(*e.g.*, *Hjelm et al., 2001*; *Persson & Brönmark, 2002*; *Svanbäck et al., 2008*), while 20 studies used published equations, mostly to determine the contribution of macroinvertebrates to fish diets (*e.g.*, *Persson, 1997*; *Byström et al., 2004*). For more information, see Table S2.

The database review of the composition of the families and genera surveyed showed some, but limited, taxonomical overlapping between the species lists of Portugal and Sweden, and a higher biodiversity at the species level in Portugal than in Sweden for most genera (see Table 3). The genus *Asellus* has only one species in both countries. The comparison of the genus *Caenis* was not possible since the data at the species level were not available for Portugal. Species diversity was only higher in Sweden than in Portugal for two genera, the genus *Cloeon* from the Ephemeroptera order and genus *Aeshna* from the Odonata. Among the taxa identified at the family level, the family Coenagrionidae has five genera occurring in Portugal and six genera in Sweden, of which four genera are shared between the two countries. For the family Dytiscidae, there are 26 genera confirmed in Portugal and 16 genera in Sweden, of which 11 genera are shared between the two countries.

### Length-mass relationships

Overall, the models explained a high amount of variance in the mass data in all taxa (range of $R^2_m = 0.80$–$0.95$), except in the genus *Caenis* ($R^2_m = 0.63$; see Table 4). The statistical comparison of the equations obtained for each taxon in Portugal and Sweden revealed similar length-mass relationships for only one out of the eight taxa compared. The potential transferability of the length-mass relationships of the genus *Caenis* was high between the two study areas, as the variable country had no significant effect on the curves obtained ($P = 0.172$; Table 4, Fig. 2). However, we found significantly different country-specific equations for the genera *Cloeon*, *Sympetrum* and *Notonecta*, and for the

**Table 4 Summary of the LMMs testing for the effect of body length, country (Portugal *vs.* Sweden) and their interaction on the body mass of each pond macroinvertebrate taxon.**

| Order | Family | Genus | Length | | Country | | Length × Country | | |
|---|---|---|---|---|---|---|---|---|---|
| | | | F | *p* | F | *p* | F | *p* | R² |
| Isopoda | Asellidae | *Asellus* | 4,086.00 | **<0.001** | – | – | – | – | 0.95 |
| Ephemeroptera | Baetidae | *Cloeon* | 807.24 | **<0.001** | 91.98 | **<0.001** | 85.07 | **<0.001** | 0.82 |
| | Caenidae | *Caenis* | 80.18 | **<0.001** | 1.99 | 0.172 | 0.010 | 0.934 | 0.63 |
| Odonata | Coenagrionidae | | 772.62 | **<0.001** | 17.07 | **0.002** | 16.62 | **0.002** | 0.88 |
| | Aeshnidae | | 661.16 | **<0.001** | 66.22 | **<0.001** | 1.19 | 0.279 | 0.95 |
| | Libellulidae | *Sympetrum* | 711.47 | **<0.001** | 24.99 | **<0.001** | 38.91 | **<0.001** | 0.91 |
| Hemiptera | Corixidae | | 520.41 | **<0.001** | 18.66 | **<0.001** | 8.30 | **0.005** | 0.90 |
| | Naucoridae | *Naucoris* | 315.84 | **<0.001** | – | – | – | – | 0.80 |
| | Notonectidae | *Notonecta* | 824.53 | **<0.001** | 10.30 | **0.007** | 0.95 | 0.363 | 0.93 |
| Coleoptera | Dytiscidae | | 67.17 | **<0.001** | 7.09 | **0.022** | 5.73 | **0.040** | 0.93 |

**Note:**
Test statistics are displayed with corrected degrees of freedom using Satterthwaite's method (F), along with associated probabilities (p) and marginal coefficients of determination (R²). *P*-values significant at α = 0.05 are shown in boldface.

families Aeshnidae, Coenagrionidae, Corixidae and Dytiscidae ($P < 0.05$ in all cases; Table 4, Fig. 2), that indicate a low potential transferability of the equations obtained for these taxa between the two regions. We did not find general consistent trends for the country-specific effect. The slope of the length-mass equations obtained for the genera *Sympetrum* and *Notonecta* and for the families Aeshnidae and Corixidae was steeper for Portugal than for Sweden (Fig. 2). However, the slope of the length-mass equation obtained for the family Dytiscidae was steeper for Sweden than for Portugal (Fig. 2). Similarly, the slope of the length-mass equations obtained for the genus *Cloeon* and the family Coenagrionidae was steeper for Sweden, but in these cases the equations for the two countries crossed, respectively, at the body length of 8.6 and 16.2 mm (Fig. 2).

We were able to find published length-mass relationships in the literature, from either Portugal or Sweden, for a single taxon. In their study, *Perkins et al. (2010)* report a length-mass relationship for *Asellus aquaticus* using individuals collected in Umeä, Northern Sweden. The equation developed in our study closely matches the equation obtained by *Perkins et al. (2010)*, especially in the smallest body size classes (Fig. 3). For larger body size classes, the equation published by *Perkins et al. (2010)* showed a steeper slope, being essentially parallel to the equation developed in this study and potentially transferrable upon application of a constant correction factor.

## DISCUSSION

Our literature overview highlights a lack of published length-mass relationships for the taxa assessed here. Our work greatly increased the number of length-mass relationships available for macroinvertebrate taxa from Southern and Northern Europe. We developed 18 new length-mass relationships for 10 genera and two families of pond

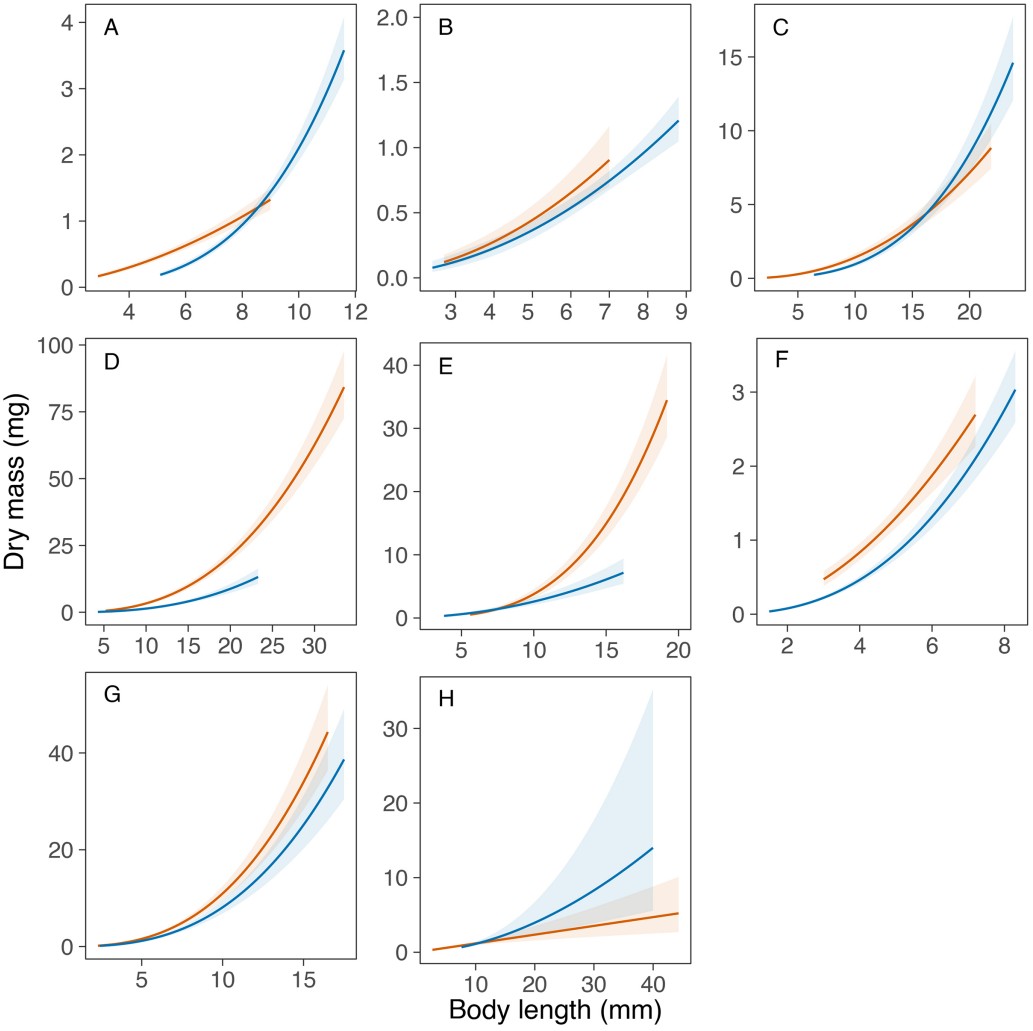

**Figure 2 Comparison of the developed length-mass relationships between Portugal and Sweden.**
Comparison of the length-mass relationships developed for the orders Ephemeroptera ((A) *Cloeon* and (B) *Caenis* genera), Odonata ((C) genus *Sympetrum*, and families (D) Coenagrionidae and (E) Aeshnidae), Hemiptera ((F) genus *Notonecta*, and (G) family Corixidae) and Coleoptera ((H) family Dytiscidae). Lines with colour bands = mean fit ± 95% confidence intervals for each taxon in Portugal (red) and Sweden (blue). 

macroinvertebrates from Portugal and Sweden. The equations obtained are the first length-mass relationships available from these two countries for 11 out of the 12 taxa assessed in this study, constituting important tools to estimate macroinvertebrate biomass and assess secondary production in pond habitats. We demonstrated that the equations obtained for each taxon are generally steeper in Portugal and, as such, are not transferrable between the two study areas in Southern and Northern Europe. In contrast, the comparison of the equation obtained for *A. aquaticus* in this study with another equation published for this taxon in Sweden showed a high transferability potential. We uncovered interesting trends in the literature overview of the studies conducted in Sweden, showing

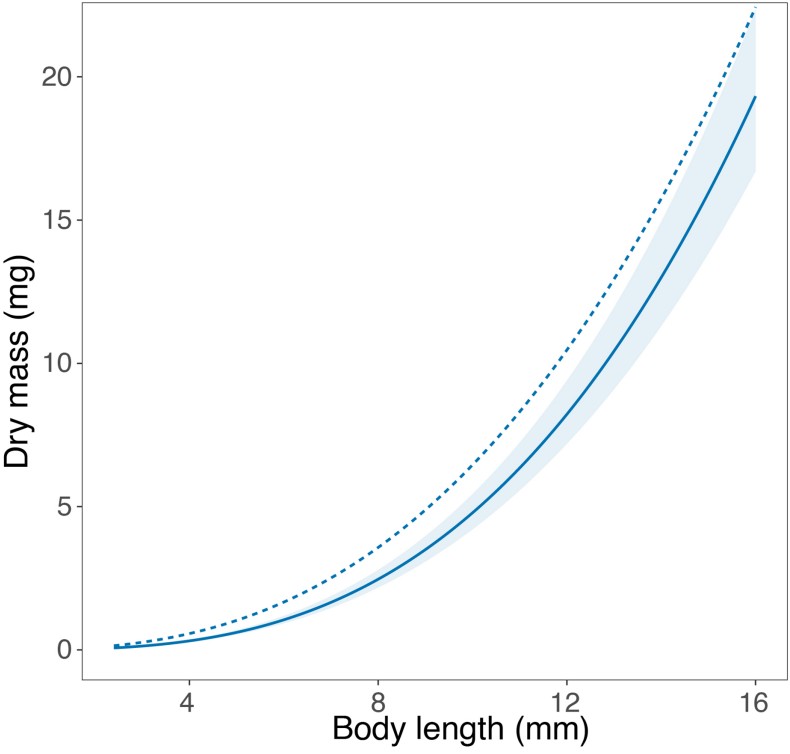

**Figure 3 Comparison of the length-mass relationships for genus *Asellus*.** Comparison of the length-mass relationship developed for the genus *Asellus* (order Isopoda) in this study with individuals from Trelleborg in Southern Sweden (solid line ± 95% confidence interval), and in the study by *Perkins et al. (2010)* with individuals from Umeå in Northern Sweden (dashed line).

that despite not publishing the equations developed, authors in this country frequently borrowed equations from literature.

## Literature review

The literature was scarce and patchy, indicating that the study of size-mass relationships in aquatic macroinvertebrates is still poorly developed, with equations available for only a restricted group of taxa and based on a multitude of size measurements across studies. Overall, we found a total of 40 studies developing length-mass relationships for the two countries. Interestingly, the equations were published in 18 out of the 20 studies found for Portugal, but only in three out of the 20 studies found for Sweden. Notably, borrowing equations from the literature was more common for studies conducted in Sweden ($n = 21$) than in Portugal ($n = 6$). This disparity may be related to the general focus of the studies, as Portuguese studies focused on spatial and temporal variation in invertebrate biomass, while Swedish studies focused on the contribution of invertebrates to fish diets. Except for the study by *Marques et al. (2003)*, we were unable to find any studies comparing the length-mass relationships developed with those published for other countries. These authors compared the size-frequency distribution, growth and biomass of populations of
one Amphipod species between Portugal, Tunisia and Italy, but did not compare the size-mass relationships.

The low number of studies with published length-mass relationships limits our ability to easily and accurately obtain biomass estimates for aquatic macroinvertebrates, forcing authors to develop their own equations. As an alternative to this time-consuming task, authors often borrow published equations from the literature, often developed in different geographic and climate regions. However, this practice can lead to gross errors in the determination of biomass estimates, with important consequences for the study of secondary production in aquatic habitats. Our review underscores the need to expand the array of readily available length-mass relationships in the literature and advocates for the publishing of the equations developed as a standard practice.

Our research on the similarity in the composition of the families and genera surveyed showed a small overlap between the species of the families Dytiscidae and Coenagrionidae in Portugal and Sweden, as could be expected at the family level. This was especially true for Dytiscidae, which is an extraordinarily big and morphologically and ecologically diverse family. On the contrary, the genera *Cloeon* and *Notonecta* showed a substantial overlap between the species occurring in the two countries, with all species of the genus *Cloeon* in Portugal and all species of the genus *Notonecta* in Sweden being shared with the other country. Similarly, most of the genus *Sympetrum* species in Sweden are shared with Portugal. However, despite the large overlap in the species pools, the length-mass equations of *Cleon* and *Sympetrum* differed between the two countries. Analysing the species overlapping for the genus *Caenis*, the only taxon with transferable equations between countries, could inform on the contribution of interspecific variation to our results, but unfortunately there are no available data for the genus *Caenis* in Portugal. Nevertheless, despite potential issues with intra- and inter-specific variation, our results indicate that the transferability of length-mass relationships is difficult to predict, even when there is a large overlap in the species composing a genus in the two countries (*e.g.*, *Cleon* and *Sympetrum*). This, suggests that other factors may determine the transferability of length-mass equations, such as local environmental conditions and climate. However, future studies at lower taxonomic levels of determination could provide clarity on the contribution of intra- and interspecific variation to the transferability of length-mass equations.

## Transferable length-mass relationships

The transferability between the equations developed for each taxon in Southern and Northern Europe was low. The genus *Caenis* was the only taxon with transferable length-mass equations between the two countries. High transferability of length-mass relationships in aquatic macroinvertebrates has been previously documented in Europe by *Rosati, Barbone & Basset (2012)*, who showed invariance in the length-mass regressions obtained for macroinvertebrate taxa in the Danube delta and in Southern Italy. These authors argued that length-mass relationships are transferrable between ecoregions and can be used to estimate population biomass, provided that the environmental conditions are similar. Here, we showed that, although infrequently, length-mass relationships may be

transferrable between ecoregions with different environmental conditions, suggesting that the potential transferability of length-mass relationships is taxon-dependent and difficult to predict. For instance, length-mass equations in the order Ephemeroptera were transferrable in the genus *Caenis*, but not in the genus *Cloeon*.

To our knowledge, only a single study reported a length-mass relationship for any of the macroinvertebrate taxa assessed here. *Perkins et al. (2010)* developed an equation for *Asellus aquaticus* (order Amphipoda) from Sweden using individuals collected in Umeä ($n$ = 50), while we used individuals collected in Trelleborg ($n$ = 211). Despite the lack of overlap between the published equation and the 95% confidence interval of the equation developed in this study, the graphical comparison shows very similar length-mass relationships for the two populations of *A. aquaticus*, especially at small body sizes. The similar intercept and the parallel slope of the two curves suggest a high potential transferability between the two equations. The average mass at length is slightly higher in the equation developed by *Perkins et al. (2010)*, which may reflect the differences in the climatic conditions at the collection sites in the two studies, separated by a large geographical distance (ca. 1,000 km). Umeä is located in Northern Sweden (63°N) and experiences colder temperatures for longer periods with shorter growing seasons than Trelleborg, which is located in Southern Sweden (55°N). The seemingly high transferability of the length-mass relationships in *A. aquaticus* may be rooted in the ecology of this species, which is known to inhabit a wide range of freshwater ecosystems, including streams, rivers, ponds and lakes, and typically has two complete generations per year (*Needham, 1938*; *Lafuente et al., 2021*). This broad ecological tolerance suggests that environmental conditions may not impose strong constraints to its life history, leading to a comparable allometry across sites. Furthermore, *Rosati, Barbone & Basset (2012)* showed the length-mass relationships of *A. asellus* to be similar across a large biogeographical scale within the same category of habitats. Hence, the stable conditions of lentic ecosystems may lead to body mass stability, reducing variation in the length-mass relationships of *A. asellus* and explaining the seemingly high transferability of the length-mass equations in this taxon.

## Non-transferable length-mass relationships

The length-mass relationships obtained were not transferrable between the two countries for seven out of the eight macroinvertebrate taxa compared in this study. Namely, the families Aeshnidae, Coenagrionidae, Corixidae and Dytiscidae, and the genera *Cloeon*, *Sympetrum* and *Notonecta*. As shown in Fig. 1, the non-transferability of the length-mass equations obtained was caused either by a different intercept (family Coenagrionidae), a different slope (family Aeshnidae and genus *Notonecta*) or a combination of both (genera *Cloeon* and *Sympetrum*; families Corixidae and Dytiscidae). This was expected and may be driven by many of factors, operating at different levels and with varying importance, that can alter length-mass relationships. While this study did not assess the effects of any specific driver, it illustrates well the potential variation in length-mass relationships within the same taxonomic group that may have been driven by different environmental

 

conditions in Southern and Northern Europe, as a consequence of the large geographic distance and latitudinal cline between the two study areas.

Studies comparing length-mass relationships of freshwater macroinvertebrate taxa at the familial level have shown high variability in model predictions for different geographic locations (*Johnston & Cunjak, 1999*; *Hajiesmaeili, Ayyoubzadeh & Abdoli, 2019*). Furthermore, environmental conditions, such as water temperature (*Schröder, 1987*) and chemistry (*Meyer, 1989*; *Basset, 1993*; *Burgherr & Meyer, 1997*), as well as food availability (*Gee, 1988*; *Basset & Glazier, 1995*) and trophic interactions (*Basset & Rossi, 1990*; *Hajiesmaeili, Ayyoubzadeh & Abdoli, 2019*) can cause intra-specific variability within instar dry mass, sex, life-stage and growth rates, constituting important drivers of changes in length-mass relationships of freshwater macroinvertebrates (*Benke et al., 1999*; *Giustini et al., 2008*; *Méthot et al., 2012*; *Hajiesmaeili, Ayyoubzadeh & Abdoli, 2019*). Curiously, mass at length was more frequently higher in Portugal (families Aeshnidae and Corixidae, and genus *Sympetrum* and *Notonecta*), but this trend in the country-specific effect was not observed systematically across all taxa. Following the temperature-size rule (*Atkinson, 1995*), the higher temperatures in Portugal should lead to smaller adult or metamorphic body sizes, but higher temperatures also accelerate growth and developmental rates (*Dallas & Ross-Gillespie, 2015*). Thus, we hypothesize that at the same body size, individuals from Portugal may be at more advanced developmental stages, which would explain the higher mass at length observed for most of the taxa surveyed. This is supported by the change in the steepness of the slopes of the length-mass relationships, that typically increases with size, suggesting that the last stages of development are characterized by a sharper increase in mass than in size.

Despite the effects of the geographic cline in the growth and life cycles on the length-mass relationships of aquatic macroinvertebrates (*Marques et al., 2003*), it is often possible to apply specific correction factors to increase their transferability. For example, the smearing correction factor used by *Mährlein et al. (2016)* successfully removed the bias introduced by the log transformation of mass in the length-mass relationships developed for macroinvertebrates found in the littoral zone of different lakes, leading to a good transferability of the equations between geographical regions. Such correction factors might apply to the family Aeshnidae ($DM_{Portugal} = DM_{Sweden} \times \exp(0.881)$) and the genus *Notonecta* ($DM_{Portugal} = DM_{Sweden} \times \exp(0.302)$), where the estimated length-mass relationships were parallel. However, the length-by-country interactions in our models (Table 4) prevent the application of a constant correction factor to improve the transferability of the equations developed for the families Coenagrionidae, Libellulidae, Corixidae and Dytiscidae and the genus *Cloeon*.

Our findings underscore the recommendations of previous studies to avoid the use of size-mass relationships obtained in study areas that are not in close geographic proximity, particularly at the edges of species' distribution ranges (*Gowing & Recher, 1985*; *González, Basaguren & Pozo, 2002*; *Mährlein et al., 2016*). However, our results suggest that at least for some taxa the size range of the individuals used to obtain the equations may determine the potential transferability of length-mass relationships. Our data shows variation with size in the error of the biomass estimates obtained through the application of the equations
developed for Portugal and Sweden, which increased with size in the taxa for which length-mass equations had a similar intercept but a different slope (*e.g.*, family Aeshnidae, and genera *Sympetrum* and *Notonecta*). This suggests that the potential error in biomass estimates obtained through the application of equations borrowed from the literature may be lower at smaller sizes, even if the equations are not transferrable across the whole size range.

Length-mass relationships are used to calculate secondary biomass and production and can be used for estimation of life history traits (*Resh, 1979*; *Cressa, 1999*; *Walther et al., 2006*; *Martin, Proulx & Magnan, 2014*). The use of inappropriate length-mass relationships can thus lead to biased estimations of energy flow in aquatic or aquatic-terrestrial food webs, population dynamics and fitness, flux of emerging species and deposition to land, as a source of transport energy and nutrients from water to land. For example, *Dekanová (2014)* detected significant over- and underestimation of the biomass of three macroinvertebrate species when comparing self-developed and borrowed length-mass relationships from different geographical regions.

## Methodological factors influencing the estimation of length-mass relationships

The development of length-mass equations is prone to methodological biases (*Meyer, Peterson & Whiles, 2011*). The field sampling, drying, measuring, and preservation methods, together with the gut contents of the individuals, the range of body lengths measured, and the sample size greatly determine the accuracy of length-mass relationships (*Morin & Nadon, 1991*; *Johnston & Cunjak, 1999*; *Dekanová et al., 2023*; *Mocq, Dekanová & Boukal, 2024*). In our study, these methodological biases are unlikely to be responsible for differences in the length-mass relationships obtained, since they were developed using the same methods of collection, sample treatment and mathematical modelling. Thus, the low transferability of the length-mass relationships obtained at the family level, and to a lesser extent at the genus level, may stem from undesirable heterogeneity in the data (*Cressa, 1999*). It is generally recommended to determine length-mass relationships at lower taxonomic levels (see *Poepperl, 1998*; *Miyasaka et al., 2008*), and numerous studies show that the accuracy of biomass estimates obtained for aquatic insects increases with taxonomic specificity (*e.g.*, *Smock, 1980*; *Benke et al., 1999*; *Sabo, Bastow & Power, 2002*). The grouping of individuals from different taxa may, at least partially, explain the large difference and high variation in the model predictions leading to the non-transferability of most of the length-mass relationships determined here. This issue is best exemplified at the family level in Dytiscidae, for which the large confidence intervals of the equations result from the grouping of two (Portugal) and three genera (Sweden) with substantially different morphology and size ranges. In contrast, the greater similarity in morphology of the genera composing the families Coenagrionidae and Corixidae lead to smaller differences in the equations obtained, adding further support to this argument.

## CONCLUSIONS

Our work greatly expanded the number of length-mass relationships available in the literature for common pond macroinvertebrates from Southern and Northern Europe. The non-transferability of the equations obtained for comparable taxa in Portugal and Sweden suggests that the environmental conditions in the two geographic regions may affect the length-mass relationships of pond macroinvertebrates. However, even though the slope of the length-mass equations was generally steeper for Portugal, that was not always the case. The difficulty in identifying clear patterns of variation highlights the need for further quantitative studies across different biogeographic regions that assess the effects of environmental drivers on size-mass relationships. Our study provides evidence that length-mass relationships can change drastically between different geographic and climatic regions, but it is based on a restrict number of ponds within a small distance that were sampled once during the growing season. Studies covering broader geographical areas and longer timeframes are needed to include intra-specific variation and develop length-mass equations over wider size ranges and draw stronger conclusions.

Our research stresses the issues with borrowing published length-mass equations from the literature, but we provide guidelines to minimize the biases potentially introduced by this practice. When the use of self-developed length-mass relationships is impossible or impractical, borrowing published equations should be done at the lowest taxonomic level and for individuals within the size range used in their development. Furthermore, equations should be taken preferably from the same geographic region or in regions with similar environmental conditions. In addition to this, we believe necessary to make a concerted effort to adopt standard methodologies, as those described in the Sample processing section of this study, to facilitate the comparison of equations between studies. Following this, we stress the importance of publishing the equations developed as a standard practice, as it expands the array of available length-mass relationships in the literature, raising their value as tools to assess secondary production in other studies. This alone could have a significant impact on research by saving time and resources in the development of already known but unpublished equations. Finally, we believe that increased collaborative research at an international level could boost the availability of length-mass equations for general use, for example through the creation of a repository.

## ACKNOWLEDGEMENTS

We thank Rui Rebelo and Diane Srivastava for their help in the field work and specimen collection.

### Funding

This research was funded by Fundação para a Ciência e a Tecnologia, I.P./MCTES through national funds (PIDDAC)—PTDC/BIA-BMA/1893/2020 (DOI: 10.54499/PTDC/BIA-BMA/1893/2020). Bruno M Carreira was supported by Fundação para a Ciência e a Tecnologia, I.P./MCTES through an individual contract awarded under the Scientific

Employment Stimulus—CEEIND/02435/2018 (DOI: 10.54499/CEECIND/02435/2018/
CP1534/CT0007). Vladimíra Dekanová benefited from a scholarship awarded by the
National Scholarship Programme of the Slovak Republic. The funders had no role in study
design, data collection and analysis, decision to publish, or preparation of the manuscript.

### Grant Disclosures

The following grant information was disclosed by the authors:
Fundação para a Ciência e a Tecnologia, I.P./MCTES through National Funds (PIDDAC):
PTDC/BIA-BMA/1893/2020.
Fundação para a Ciência e a Tecnologia, I.P./MCTES through an Individual Contract
Awarded under the Scientific Employment Stimulus: CEEIND/02435/2018.
National Scholarship Programme of the Slovak Republic.

### Competing Interests

The authors declare that they have no competing interests.

### Author Contributions

- Vladimíra Dekanová conceived and designed the experiments, performed the experiments, analyzed the data, prepared figures and/or tables, authored or reviewed drafts of the article, and approved the final draft.
- Marek Svitok analyzed the data, prepared figures and/or tables, authored or reviewed drafts of the article, and approved the final draft.
- Sara Bento performed the experiments, prepared figures and/or tables, authored or reviewed drafts of the article, and approved the final draft.
- João Caramelo performed the experiments, prepared figures and/or tables, authored or reviewed drafts of the article, and approved the final draft.
- Pedro Peixe e Sousa performed the experiments, prepared figures and/or tables, authored or reviewed drafts of the article, and approved the final draft.
- Bruno M. Carreira conceived and designed the experiments, performed the experiments, analyzed the data, prepared figures and/or tables, authored or reviewed drafts of the article, and approved the final draft.

### Data Availability

    The raw measurements are available in the Supplemental File.

### Supplemental Information

Supplemental information for this article can be found online at http://dx.doi.org/10.7717/
peerj.18576#supplemental-information.

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
