# Peer review of "Length-mass relationships of pond macroinvertebrates do not hold between Southern and Northern Europe"

_PeerJ, doi:10.7717/peerj.18576_

## Round 0.1 · original submission · Minor Revisions

All reviewers agree your paper has merit, but have raised a few concerns, which I would like to see addressed in your revisions, these include:

[1] The need to provide practical guidance/advice regarding how researchers should work with length-mass relationships given the shortcomings you have indicated;
[2] Provide more justification for the use of specimens not ID'd to species, given the possibility that different species may occur in southern and northern Europe which may confound comparisons;
[3] Consider providing more details of the statistical analysis in the supplementary material as recommended by reviewer 2.

Reviewer 1 ·

Basic reporting

This manuscript makes a strong argument for increasing the number of length-mass relationships for aquatic macroinvertebrates. The authors present a literature review to establish the scope of available length-mass relationships, generate their own length-mass relationships from empirical data, and then compare the models among matched taxa in different locations and/or the literature. The title of the work and the abstract state the results assertively. The results are not surprising, but they are important. The authors verify that common assumptions regarding the transferability of length-mass relationships are incorrect or at least more complicated than previously acknowledged, yet; the authors offer little guidance for overcoming this obstacle beyond additional studies. Recommending collaboration and method standardization may be within the scope this manuscript to propose.

Overall, the manuscript is generally well-written and well-organized, but there are a few instances where language could be clarified.

Line 29: Begin the sentence with “A” instead of “The”. Delete “collected”.
Line 30: I think this statement would read more clearly if it were separated into two sentences.
Line 30: Almost no details about the experimental design for the development of the length-mass relationships are provided in the abstract. Perhaps some mention of sampling effort or particulars of the analysis could be included in the abstract.
Line 33: Begin the sentence with “A” instead of “The”. Change “but” to “except”. Change the word order to “potential transferability”.
Line 47: What is meant by the phrase “the dynamic of aquatic ecosystems”? This statement is too vague.
Line 59: Replace “utilizing” with “using”.
Line 73: This statement could be clarified. I think the phrase “As a result” could be eliminated by adapting the clause “A consequence of this intraspecific variation” to take its’ place.
Line 77: Replace “at hand” with “actual”.
Line 81: Change “transferability potential” to “potential transferability” here and throughout the paper.
Line 155: I believe invertebrate should be plural.
Line 160: Replace “in” with “with”.
Line 238: “potential transferability”
Line 241: “potential transferability”
Line 252: This statement could be more concise. Is the phrase “Among all the macroinvertebrate taxa surveyed in this study” necessary? Could it be shortened or eliminated?
Line 262: Replace the “the” after “highlights” with “a”. The second part of the sentence (beginning with “uncovered”) should be a separate sentence and might even be shifted in the overall organization of this paragraph.
Line 272: “potential transferability”
Line 281 and Line 282: Both of these sentences start with “Interestingly”. Perhaps consider changing one of these instances?
Line 293: Replace “many times” with “often”.
Line 303: This sentence could be more concise.
Line 315: Add “only” before “a single study...”.
Line 321: “potential transferability”
Line 325: Change “and” to “with” before “shorter”.
Line 336: Remove “to an extent”. Change “multitude” to “many”.
Line 338: Remove “the” before “different”.
Line 345: Check parentheses throughout this sentence.
Line 357: Do you have a reference for this statement?
Line 383: Do you think this could be because there are physiological constraints on the starting size of aquatic invertebrates?

Experimental design

While the study design seems adequate to address the primary question presented in the paper, I am not sure that the sampling would be spatially or temporally adequate to ensure that the length mass relationships generated would meet the potential transferability criteria desired even within the same geographical area (i.e., country). The comparison of sites within Sweden seems to validate this concern. So, while one objective, to compare length-mass relationships over a large geographic area was achieved. I am not convinced that their sampling at only a few sites relatively close together in one season would be sufficient to capture the full range of sizes as well as intraspecific variation would be any more useful to other researchers as the other published relationships the authors appear to be critiquing. The authors do somewhat thoroughly address confounding factors and inadequacies in the discussion. However, the authors could perhaps offer some recommendations for how more robust relationships could be established moving forward or how to verify these relationships over space. As mentioned previously the authors verify a problem but offer only limited solutions.

Validity of the findings

In my opinion the data analysis was appropriate and sufficient details provided where the results could be reasonably replicated except for how the size classes were determined. This point could be clarified. Also, in the results section, the authors could more thoroughly compare the taxa among the two locations. I presume that while families and genera are distributed widely enough across this portion of Europe to be represented in the invertebrate community the species composition in each location is likely to differ. The authors do discuss taxonomic resolution in the discussion, but I would like to have a more thorough comparison in the results, if known. For example, if a genus occurs in both locations are the length mass relationships a compliment of 5 potential species in Portugal but only 2 in Sweden?

Additional comments

The authors of this work provide empirical evidence verifying the inadequacy of our current understanding of length-mass relationships in aquatic macroinvertebrate for studies of secondary production and other aspects of aquatic ecology. Yet, the models the authors provide in this study may have limited scope due to reasonable, practical limits in the field and laboratory. Before this work is published, I think the authors could provide some additional commentary on reasonable spatial and temporal scope when developing length-mass relationships as well as ways to simplify and / or standardize efforts to develop these models since the need to develop additional models is the overall implication of their study.

Reviewer 2 ·

Basic reporting

Journal: Peerj
MS: 104053v1
Title: Length-mass relationships of pond macroinvertebrates do not hold between Southern and Northern Europe

General comments

This manuscript describes length-mass relationships of pond macroinvertebrates in two European regions. The review of published length-mass relationships from two countries at latitudinal extremes (Portugal and Sweden) highlights potential significant changes between geographic and climatic regions. The expansion of available equations for pond macroinvertebrates in Europe may be valuable for future research and comparative studies. The study is interesting, with an adequate sample size and duration, and the results are clear. However, I have some concerns regarding this manuscript.

For PeerJ readers, these taxa may not be familiar. Therefore, including basic biological and ecological characteristics would help improve understanding. Please consider adding some paragraphs addressing this.

There are some minor grammatical, formatting, and paragraphing issues. Despite these, the manuscript is a valuable contribution to the literature. Specific comments are listed below.

Experimental design

Specific comments

L. 67-69: Please specify the species/taxa rather than using "most species."

M&M: Include information on animal ethics approval and authorizations if applicable.
L. 139-142: Provide the literature used for the identification and classification of families and genera.

Figure 1

The map of the two locations could provide more detailed information about the pond areas and their surroundings.

Validity of the findings

Specific comments

A. aquaticus is mentioned as showing a high transferability potential. You could expand on the ecological or physiological reasons behind this observation.

L. 282: The sentence starting with “Interestingly, borrowing published equations from the literature...” could be more concise. Additionally, discuss briefly whether borrowing equations is acceptable in certain contexts or if it should generally be avoided.

L. 329: The argument for non-transferability of length-mass equations across regions, especially for certain taxa, could be expanded. Consider discussing the ecological significance of these differences, such as how they might impact studies on food webs, nutrient cycling, or other ecological processes.

Finally, include histograms of mass and length for each taxon, as well as homoscedasticity, normality, and residuals plots in the supplementary materials.

Annotated reviews are not available for download in order to protect the identity of reviewers who chose to remain anonymous.

Reviewer 3 ·

Basic reporting

No comment

Experimental design

No comment

Validity of the findings

This analysis investigating the role of context in the usefulness of length-mass equations is an important idea for anyone attempting to calculate biomass estimates of aquatic macroinvertebrates. I found the manuscript to be well-written, and I have no issue with the statistical approach.

However, I had one major concern with the paper that I feel needs to be addressed before publication, given the implications that are claimed. Almost all of the literature reviewed for taxa in Portugal and Sweden gave equations at the species level-- if your specimens were only identified to the family or genus level, how much of the variation in mass-length relationships can be attributed to location, and how much can be expected of interspecific variation? What kind of overlap in species between Sweden and Portugal exists for each taxonomic group examined (genus / family)? I agree that context likely does have an effect on physiology, but it's hard to know that location alone is driving the effect. This is mentioned in the discussion (lines 395-408), but I think that much more attention needs to be given to this aspect.

Two things could help strengthen the conclusions of your paper:
1. Return to your specimens and complete species IDs and see if there are any overlap between Sweden and Portugal, then see if the shared species show the same level of non-transferability as do the genus models. I understand this may be very difficult or impossible, but perhaps this suggestion could shape future studies
2. Present from the literature, what proportion of species in each genus / family are shared between Portugal and Sweden and therefore how much interspecific variation might be acting on the results

With regard to publications that borrow equations from different locations-- I'm assuming the species matched between locations (since the published equations were often at the species level), though the measure of variation from the present study that is applied to the "borrowing" again doesn't make it clear whether it actually applies to intraspecific geographical variation. Perhaps if you broadened your search of usage of equations, it could help make your case-- if researchers are borrowing genus- or family-level equations (not sure if that is the case) from different geographic areas, your conclusions apply.

I think, however the authors decide to adjust the paper, the main thing that would be helpful is being very clear about how much interspecific variation may exist within the dataset (i.e., how many species are in each taxonomic group in each location, and what proportion is shared between Portugal and Sweden, thus: how likely are these the same species or different species being measured?) and how prevalent the problem of is of sharing genus or family equations across different locations.

---

## Round 0.2 · accepted · Accept

Thank you for your detailed responses to all the reviewer comments. I have reviewed these in relation to the revised manuscript, and I am satisfied that there is no need for further review, and thus the manuscript can go forward for publication.